# Vortex-based soft magnetic composite with ultrastable permeability up to gigahertz frequencies

Guohua Bai [1], Jiayi Sun[1], Zhenhua Zhang[1], Xiaolian Liu[1], Sateesh Bandaru[1], Weiwei Liu[1], Zhong Li[1], Hongxia Li[1], Ningning Wang [2] & Xuefeng Zhang [1] ✉

Soft magnetic materials with stable permeability up to hundreds of megahertz (MHz) are urgently needed for integrated transformers and inductors, which are crucial in the more-than-Moore era. However, traditional frequency-stable soft magnetic ferrites suffer from low saturation magnetization and temperature instability, making them unsuitable for integrated circuits. Herein, we fabricate a frequency-stable soft magnetic composite featuring a magnetic vortex structure via cold-sintering, where ultrafine FeSiAl particles are magnetically isolated and covalently bonded by $Al_2SiO_5/SiO_2/Fe_2(MoO_4)_3$ multi-layered heterostructure. This construction results in an ultrastable permeability of 13 up to 1 gigahertz (GHz), relatively large saturation magnetization of 105 $Am^2/kg$ and low coercivity of 48 A/m, which we ascribe to the elimination of domain walls associated with almost uniform single-vortex structures, as observed by Lorentz transmission electron microscopy and reconstructed by micromagnetic simulation. Moreover, the ultimate compressive strength has been simultaneously increased up to 337.1 MPa attributed to the epitaxially grown interfaces between particles. This study deepens our understanding on the characteristics of magnetic vortices and provides alternative concept for designing integrated magnetic devices.

Soft magnetic materials with stable permeability ($\mu$) produce alternating magnetic flux under magnetic fields driven by current, and serve as the fundamental materials for basic electronic components such as inductor and transformer[1]. In the newest more-than-Moore integrated circuit packaging technologies, PCB-embedded inductors above 100 MHz are designed[2], which require soft magnetic materials with stable permeability at such high frequency. Various Fe-based magnetic alloys with good static magnetic properties, i.e. large saturation magnetization ($M_s > 200$ $Am^2/kg$) and high initial permeability ($\mu_i > 10000$) have been developed[3]. However, soft magnetic materials are subject to working under alternative field, and the most fundamental requirement is a suitable permeability at working frequency from engineering perspective. Traditional Fe-based magnetic alloys suffer disastrous permeability "collapse" above kilohertz due to

low electrical resistivity ($\rho < 10^2$ $\mu\Omega\cdot cm$) and domain wall resonance, leading to the malfunction of circuits containing the inductor. Spinel and hexagonal ferrites possess high resistivity ($\rho > 10^{12}$ $\mu\Omega\cdot cm$), and thus can serve applications up to gigahertz[4]. Nevertheless, the low $M_s$ (~50 $Am^2/kg$) is not ideal for miniaturized inductors, and their low Curie temperatures (~200 °C) induces magnetic deterioration in integrated circuits. It remains challenging to prepare magnetic materials with simultaneous frequency-stable permeability and large saturation magnetization.

Soft magnetic composites, cold-pressed from insulated metallic magnetic particles, have shown great potential in high-frequency inductors due to their simultaneously high $M_s$ and high $\rho$[5]. However, current soft magnetic composites use metallic magnetic particles with a multidomain structure[6-11], which still leads to permeability decline

[1]Institute of Advanced Magnetic Materials, College of Materials and Environmental Engineering, Hangzhou Dianzi University, Hangzhou 310012, China.
[2]Department of Electronics and Information, Hangzhou Dianzi University, Hangzhou 310012, China. ✉e-mail: zhang@hdu.edu.cn

above megahertz frequency as a result of domain wall resonance[12]. The coercivity also deteriorates (~800 A/m) compared with bulk materials (< 80 A/m)[9,13,14], due to the internal stress introduced during cold-pressing process. Moreover, traditional soft magnetic composites crack easily due to the weak interlock capacity of spherical particles and bonding strength of the metal/coating interface. The cold pressing process also inevitably damages the integrity of insulating coatings and decreases electrical resistivity of soft magnetic composites. Evidently, new magnetic topology of particles and consolidation technique should be exploited to address the drawbacks in present soft magnetic composites.

A magnetic vortex is a basic magnetic topology in micro or nano magnetic materials. It contains a curling spin structure around a central region where magnetic moments are pointing out of plane to avoid creating a singularity[15]. Magnetic vortexes have been extensively investigated by virtue of their zero magnetostatic energy, topological protection and high resonance frequency[15]. However, previous research primarily focuses on two-dimensional magnetic vortexes and their potential employment in spintronics[16]. There are few reports on three-dimensional magnetic vortex materials and their application in traditional magnetic devices, which is mostly due to the difficulties in industrial scale production and effective consolidation of ultrafine magnetic particles. Cold sintering technique, a recently reported low-temperature low-pressure consolidation process, provides the possibility of consolidating micro or nano particles[17,18]. The basic process involves the uniform wetting of powders by aqueous solution, partially dissolution of solid surfaces in the hydrothermal environment, and the precipitation of mass at particle–particle interfaces[19]. Although cold sintering technique is commonly used for fabricating advanced ceramic materials, the dissolution-precipitation process also provides the possibility of one-step insulation and consolidation of ultrafine metallic particles with moderate pressure and temperature.

Here, we report on the effective cold sintering of soft magnetic composite from ultrafine FeSiAl particles. This composite features isolated magnetic vortexes rather than domain walls, as is the case in traditional multidomain materials from magnetic structure perspective. FeSiAl (known as Sendust alloys) presents zero magneto-crystalline anisotropy and magnetostriction constant simultaneously, which meets the intrinsic requirements of magnetic vortex formation. Gas atomization and airflow classification are used to produce micro-spherical FeSiAl particles with vortex structure in tens of kilograms. During the cold sintering process, the $Al_2SiO_5/SiO_2/Fe_2(MoO_4)_3$ multilayered heterostructure eventually forms, which magnetically isolates neighboring vortexes and covalently bonds these ultrafine FeSiAl particles. This unique structure endows the composite with frequency-stable permeability, relatively large magnetization saturation, low coercivity and high ultimate compressive strength.

## Results and discussion
### Microstructure characterization
The cold sintering process is shown in Supplementary Fig. 1. Ultrafine FeSiAl spherical particles ($d_{50}$ = 2.6 μm, Supplementary Fig. 2) are mixed homogeneously with water and $(NH_4)_6Mo_7O_{24}·4H_2O$ (AMT). Afterwards, the slurry is cold sintered under 400 MPa and 250 °C for 1 h. The corresponding composite is designated as CS-AMT&$H_2O$. For comparison, cold-sintered samples from FeSiAl powders and $H_2O$ (termed as CS-$H_2O$), pure FeSiAl powders (termed as CS-powder-only), as well as traditional cold-pressed samples from phosphoric acid passivation with silicon resin binding (termed as CP-PA&SR), and with polyvinyl alcohol binding (termed as CP-PVA) are also prepared. Meanwhile, ultrafine ($d_{50}$ = 2.0 μm) carbonyl iron and FeSiCrB amorphous particles are also mixed with water and AMT, and cold-sintered under the same condition to validate the universality of vortex-based composites. The corresponding composites are designated as CS-AMT-Fe&$H_2O$ and CS-AMT-Am&$H_2O$, respectively.

The sectional view of CS-AMT&$H_2O$ composite by scanning electron microscopy (SEM) is shown in Supplementary Fig. 3, 4, which reveal highly compacted structure with low porosity. The energy dispersive spectroscopy (EDS) elemental mappings in Supplementary Fig. 3b and c indicate that Fe, Si and Al remain in the particle matrix, while Mo, Fe and O fill the inter-particle space. The microstructure of CS-AMT&$H_2O$ composite is investigated by transmission electron microscopy (TEM) in Fig. 1. For a FeSiAl particle with diameter of 2.0 μm in Fig. 1a, magnetic vortex structure is observed in Fig. 1b by differential phase contrast (DPC), which is entirely distinct from the multidomain structure in traditional soft magnetic material. Figure 1c presents the high-resolution TEM (HRTEM) image of the particle, in which (022) and (0$\bar{2}$2) planes of face-centered cubic (FCC) FeSiAl with $d$-spacing of 2.01 Å can be identified. The selected area electron diffraction (SAED) pattern in Fig. 1g confirms the monocrystal structure of the particle viewed along [100] direction. The characterizations of other particles in Supplementary Fig. 5 also demonstrate the ubiquity of monocrystal and vortex structure in the particles we adopted. Thus, the effect of grainboundary on magnetic properties can be excluded in later discussion.

Meanwhile, a distinct coating can be observed on the surface of FeSiAl particle, as indicated by white arrow in Fig. 1a. HRTEM image of the white square zone in Fig. 1d reveals that this coating layer is composed of three disparate sublayers. Enrichment of Al/Si in sublayer I, Si in sublayer II, and Mo/Fe in sublayer III are observed from the EDS mappings in Fig. 1k–p. Sublayer I is attributed to be $Al_2SiO_5$ viewed along axis of [100], which is confirmed by the diffraction spots of (022) and (040) planes in fast Fourier transform (FFT) in Fig. 1h. The interface between FeSiAl matrix and sublayer I is further investigated with spherical aberrated scanning transmission electron microscopy (STEM) in Fig. 1e, which clearly reveals the epitaxial growth of $Al_2SiO_5$ (022) plane on the FeSiAl (110) plane. Sublayer II presents no diffraction signal in the FFT image (Fig. 1i), indicating the formation of amorphous $SiO_2$. The HRTEM image in Fig. 1f shows the nanocrystalline structure of sublayer III, in which (222) and (200) planes of $Fe_2(MoO_4)_3$ with $d$-spacings of 2.95 Å and 4.70 Å are identified. The polycrystalline structure of sublayer III is also confirmed by FFT pattern with diffraction rings of $Fe_2(MoO_4)_3$ indexed in Fig. 1j. The observation of $Al_2SiO_5/SiO_2/Fe_2(MoO_4)_3$ multilayered heterostructure is in consistent with the X-ray photoelectron spectroscopy (XPS) results in Supplementary Fig. 3e, f, which prove that metallic FeSiAl particles are partially oxidized during cold sintering.

### Static and high-frequency properties of composites
We begin illustrating the static properties of vortex-based composite by exploring a single FeSiAl particle. Because it is still challenging to observe 3D magnetic structure directly, the projected DPC image for a FeSiAl particle with size of 150 nm is taken to illustrate the vortex structure (Supplementary Fig. 6). We also use micromagnetic simulation to investigate the magnetic feature of FeSiAl particle with different diameters (Fig. 2a). Theoretically, magnetic structure and coercivity of a particle depend on its geometry, intrinsic anisotropies and inter-particle interaction. For spherical FeSiAl particle, the effects of shape anisotropy, intrinsic magnetocrystalline anisotropy, as well as magnetostriction (stress anisotropy) can be excluded. It is found that for particles smaller than 50 nm, magnetic monodomain state is the most stable configuration. However, its coercivity is higher than 100 kA/m, which is not desired for soft magnetic composites. Although superparamagnetic behavior with small coercivity can be induced in monodomain nanoparticles, their magnetization will also be deteriorated due to thermal perturbation[20]. The magnetic structure transforms to vortex state from 50 nm to ~3.0 μm (Fig. 2b), and evolves to multidomain state with larger diameters. Meanwhile, the particle coercivity continues decreasing with increased diameters. We take the particle size distribution into consideration and develop a method to

estimate the theoretical coercivity of the composite (Supplementary Table 1), with a value of 91 A/m obtained. As is well known, micromagnetic simulation represents the behaviour at 0 K, and thermal perturbation could decrease coercivity. Thus, the simulated value is higher than the room-temperature value (48 A/m) of CS-AMT&$H_2O$ composite (shown as red star point in Fig. 2a).

Theoretically, magnetic exchange coupling takes place if the distance between two particles is smaller than exchange length (~10 nm). This inter-particle interaction may remarkably affect the static properties of vortex assembly. Figure 2c simulates the hysteresis loops for isolated and contacted four-particle vortex assemblies (simulation parameters are provided in method section), with coercivity of 64 A/m and 230 A/m obtained respectively. The remanence-state magnetization configuration (insets of Fig. 2c) for contacted vortex assembly illustrates that interfacial magnetic moment are exchange-coupled and aligned parallelly, which induce an effective anisotropic field and lead to larger $H_c$. On the contrary, isolated vortex facilitates magnetization reversal process and is beneficial for smaller $H_c$. It is interesting that isolated vortex assembly presents smaller slope of hysteresis loop than contacted assembly, which indicates a lower permeability. This can be explained from its higher Zeeman energy from the simulation results (Supplementary Fig. 7). Simulations of assemblies with more particles show similar results in Supplementary Fig. 8.

In experimental demonstration, we compare the hysteresis loops for CS-AMT&$H_2O$ and CS-$H_2O$ composites in Fig. 2d, from which coercivity of 48 A/m and 270 A/m are obtained, respectively. For the CS-$H_2O$ composite, only $H_2O$ is applied during the cold sintering process. FeSiAl particles are supposed to contact with each other after the evaporation of $H_2O$. The experimental values are in good consistence with simulation, signifying that the inter-particle interaction in CS-AMT&$H_2O$ can be neglected, and most FeSiAl particles act as independent magnetic vortexes. From the fracture surface shown in Fig. 2e, f, it is clear that FeSiAl particles in CS-AMT&$H_2O$ are well separated while particles in CS-$H_2O$ contact with each other (as indicated by the white arrow Fig. 2f). In the DPC image of FIB-thinned CS-AMT&$H_2O$ sample with 6 vortexes (Fig. 2g), we observed that most FeSiAl particles are magnetically separated by $Al_2SiO_5$/$SiO_2$/$Fe_2(MoO_4)_3$ multilayered heterostructure. Meanwhile, two contacted particles are also observed (Supplementary Fig. 9). At the contact point, neighboring magnetic moments are aligned parallelly to reduce the exchange energy at interface. At this circumstance, multidomain structure has lower energy (Supplementary Fig. 10), and is more stable than isolated vortex, in which neighboring magnetic moments along the domain wall are aligned antiparallelly. Electron holography characterization is also applied to confirm the vortex and multidomain structure of FeSiAl particles in Fig. 2f (Supplementary Fig. 11). Above results confirm the role of

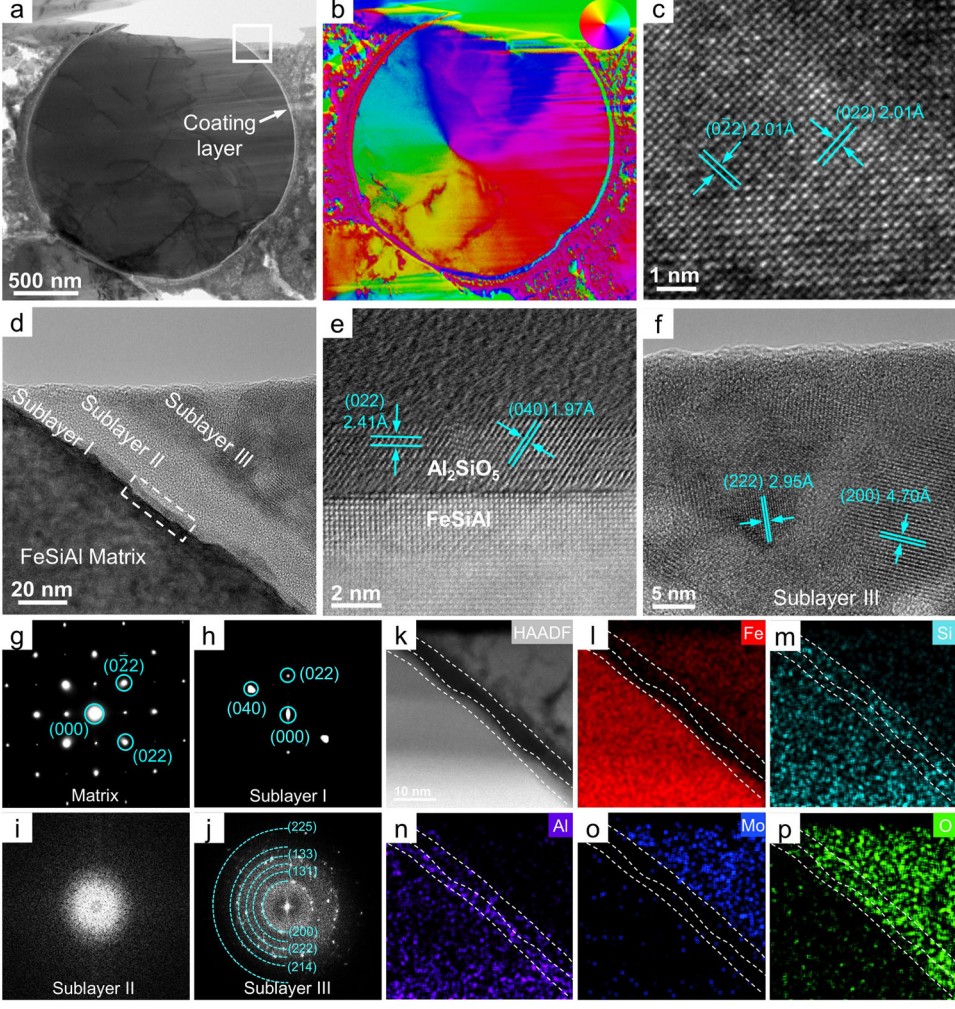

**Fig. 1 | Microstructure and domain structure characterization of CS-AMT&$H_2O$ composite. a**, **b** TEM and DPC image of the composite. **c** HRTEM image of FeSiAl particle. **d** White square region in (**a**). **e** STEM image of the dash rectangular region in (**d**). **f** HRTEM of sublayer III. **g** SAED pattern of FeSiAl matrix. **h**–**j** FFT patterns of sublayer I, II, III. **k**–**p** High angle angular dark field (HAADF) images and EDS elemental mapping of white square region in (**d**).

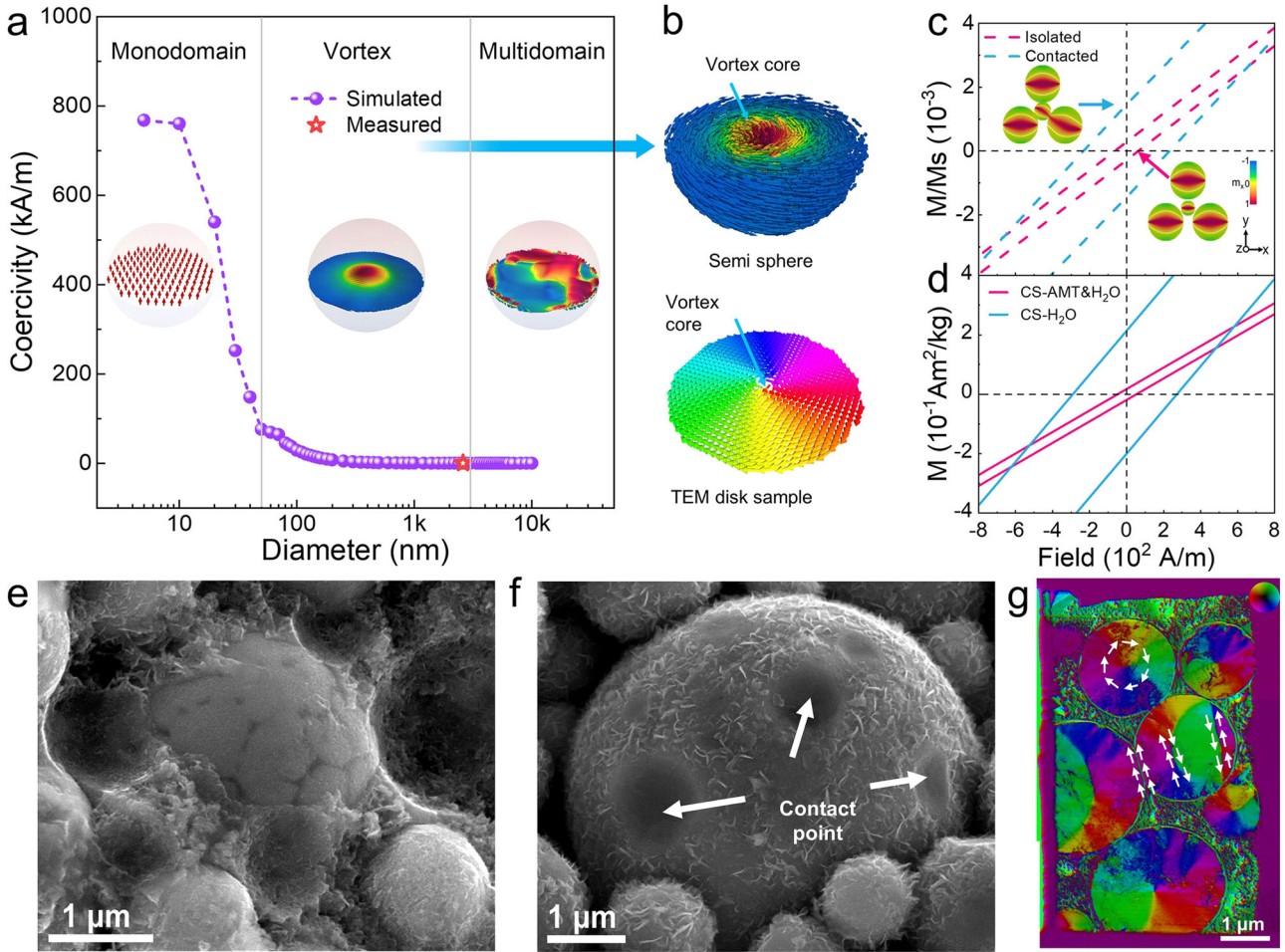

**Fig. 2 | Domain structure and static magnetic properties of FeSiAl particles.**
**a** Simulated domain structure and coercivity of FeSiAl particle with different diameters, the star point shows the experimental coercivity of CS-AMT&H$_2$O.
**b** Reconstructed semi sphere and disk vortex of FeSiAl particle with a diameter of 2.6 μm. **c** Simulated near-zero hysteresis loops for isolated and contacted four-vortex assemblies, the insets show the residual magnetization. **d** Measured near-zero hysteresis loops for CS-AMT&H$_2$O and CS-H$_2$O. **e**, **f** SEM fracture surfaces for CS-AMT&H$_2$O and CS-H$_2$O. **g** DPC image of FIB-thinned CS-AMT&H$_2$O with 6 vortexes.

continuous Al$_2$SiO$_5$/SiO$_2$/Fe$_2$(MoO$_4$)$_3$ multilayered heterostructure as magnetic isolator in realizing single-magnetic-vortex composite with reduced coercivity.

Figure 3a, b plot the permeability spectra of CS-AMT&H$_2$O composite and contrast samples. CS-AMT&H$_2$O presents an initial permeability ($\mu_i$, defined as permeability at 1 MHz) of 13 and remains almost constant up to 1 GHz. Meanwhile, CS-AMT-Fe&H$_2$O and CS-AMT-Am&H$_2$O composites present similar permeability stability. However, CS-H$_2$O and CP-PA&SR composites present drastic permeability decline around 100 MHz. In the imaginary permeability, only one resonance peak above 1 GHz can be found for CS-AMT&H$_2$O, CS-AMT-Fe&H$_2$O and CS-AMT-Am&H$_2$O composites, while additional resonance peak at 100 MHz is observed for CS-H$_2$O and CP-PA&SR counterparts, which may correspond to the domain wall resonance. In Fig. 3c, CS-AMT&H$_2$O also presents the highest qualify factor. For traditional CP-PA&SR composite, the high molding pressure (2 GPa) destroys the integrity of insulation coating (Supplementary Fig. 12), resulting in a similar frequency dependence of permeability to CS-H$_2$O. Above results confirm the crucial role of magnetic isolation on permeability stability. Figure 3d, e compare the $M_s$, $\mu_i$ and maximal stable frequency (defined as frequency of 90% $\mu_i$) of traditional magnetic composites, ferrites and our composites. We succeed in obtaining simultaneous high maximal stable frequency (up to 1 GHz), large $M_s$ (105~176 Am$^2$/kg, Supplementary Fig. 13) and high $\mu_i$ in the vortex-

based composites made from ultrafine FeSiAl, Fe and amorphous FeSiCrB particles. Above excellent magnetic properties enable the vortex-based soft magnetic composites to be perfect magnetic core materials for high-frequency inductors. Figure 3f shows the performance of PCB-embedded inductor (Supplementary Fig. 14) made from CS-AMT&H$_2$O. One can find that the inductor presents stable inductance below 1 GHz and much higher qualify factor at 100 MHz than literatures[21–26], which is of great application potential in integrated circuits.

The frequency-stable permeabilities in CS-AMT&H$_2$O, CS-AMT-Fe&H$_2$O and CS-AMT-Am&H$_2$O composites originate from the vortexes that are magnetically isolated. As predicted by Snoek[27], the permeability of magnetic material could remain stable until nature resonance frequency, which describes the uniform Larmor procession of all spins in the material. The nature resonance frequency ($f_r$) is always at GHz, and limited by the product of $M_s$ and electron gyromagnetic ratio ($\gamma$). The validation of Snoek's predication is based on idealized assumptions of perfect monocrystals, rotational magnetization process etc., but most real magnetic materials deviate from ideality, such as polycrystalline and multidomain structure. The actual permeability declines drastically at frequencies that are much below those predicted ones. A magnetically isolated vortex, where spins are aligned spirally, facilitates the magnetization rotation process. This magnetic configuration matches well with Snoek's assumption and

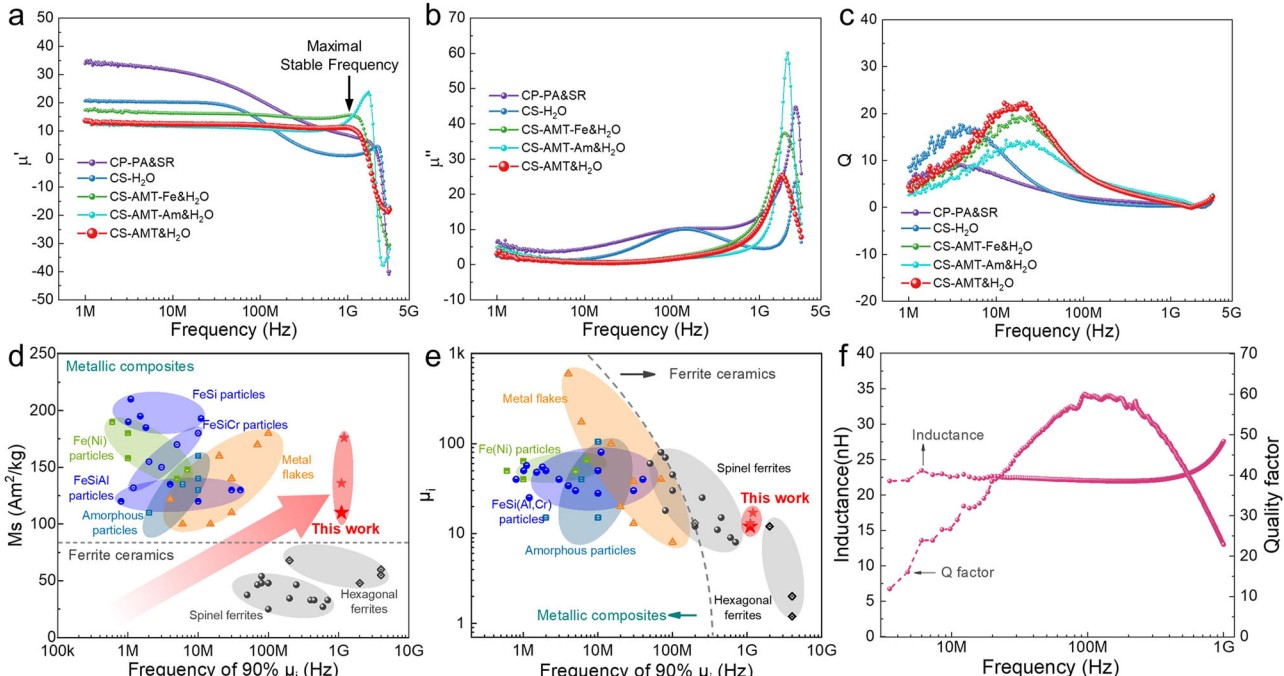

**Fig. 3 | High-frequency performance comparison. a–c** Frequency dependence of real permeability ($\mu'$), imaginary permeability ($\mu''$) and quality factor (Q) of different composites. **d, e** Comparison of $M_s$, $\mu_i$, and maximal stable frequency between vortex-based soft magnetic composites in this work and literature results[6–10,13,14,28–56]. Ferrites materials are colored by grey because of their high sintering temperature and low Curie temperature. **f** Performance of PCB-embedded inductor made from CS-AMT&$H_2O$ composite.

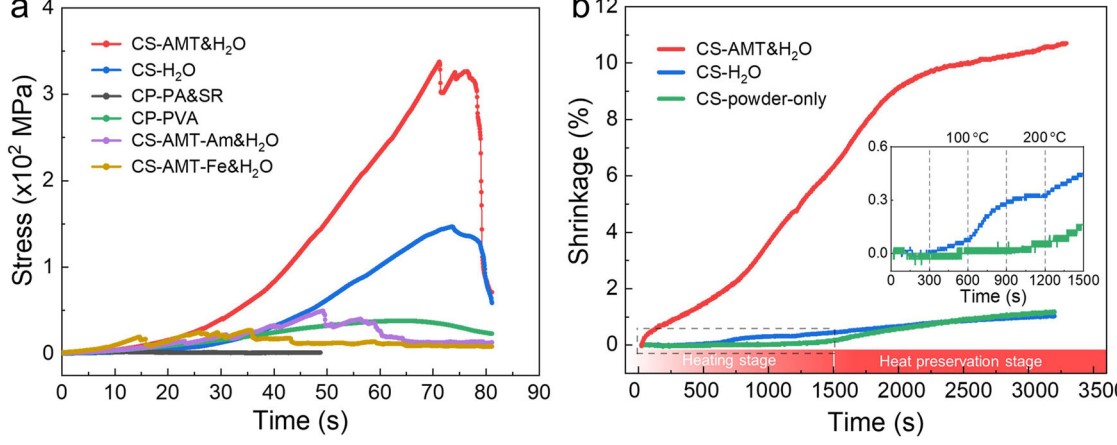

**Fig. 4 | Mechanical properties and sintering behavior of different composites. a** Compression curve comparison between cold-sintered and cold-pressed composites. **b** Shrinkage curves of cold-sintered composites CS-AMT&$H_2O$, CS-$H_2O$ and CS-powder-only. The inset shows the magnified curve for CS-$H_2O$ and CS-powder-only samples in the heating stage.

only a nature resonance peak above 1 GHz is observed. On the contrary, the magnetic vortexes in CP-PA&SR and CS-$H_2O$ composites contact with each other, which induces multidomain structure and destabilizes the permeability against increasing frequency.

## Cold sintering mechanism and mechanical strength
In practical application, high-frequency inductors are always packaged in printed circuit board by epoxy resin sealant at high pressure. Thus, adequate mechanical strength should be instilled in the inductor materials. Figure 4a compares the compression curve and mechanical strength of cold-sintered composites with those of traditional cold-pressed composites. It is observed that CS-AMT&$H_2O$ composite presents an ultimate compressive strength of 337.1 MPa, much higher than

other samples. This result demonstrates that CS-AMT&$H_2O$ composite is consolidated efficaciously even though the pressure is much lower than that applied in cold-pressed samples (400 MPa vs. 2 GPa). The effective consolidation of ultrafine particles has been a bottleneck in powder metallurgy. To unveil the cold sintering mechanism of CS-AMT&$H_2O$ composite, the shrinkage curve of CS-AMT&$H_2O$, CS-$H_2O$ and CS-powder-only composites during cold sintering process are recorded in Fig. 4b. In the case of CS-powder-only composite (indicated by green line), no obvious shrinkage is observed until 200 °C, after which creep deformation occurs, resulting in a total shrinkage of 1.19% at the end of heat preservation stage. For CS-$H_2O$ composite (indicated by blue line), the volume remains almost unchanged below the boiling point of water (100 °C). The water evaporation continues

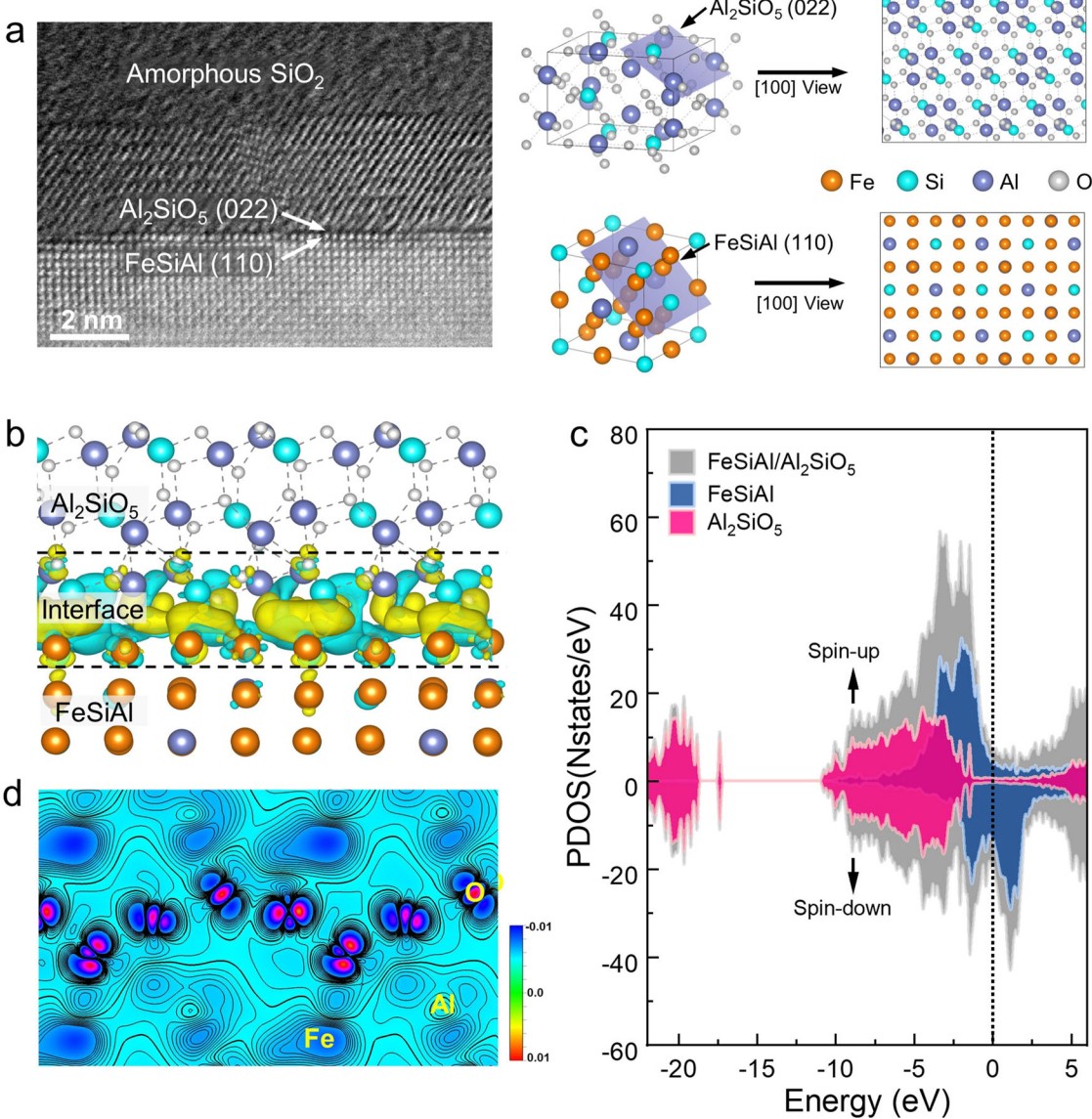

**Fig. 5 | First principle investigation of the interface. a** STEM image and crystal structure of $(110)_{FeSiAl}/(022)_{Al_2SiO_5}$ interface. **b** Three-dimensional charge density difference of interface. **c** PDOS of FeSiAl, Al$_2$SiO$_5$ and interface. **d** Two-dimensional charge density difference of interface.

until the temperature reaches 200 °C, leading to a shrinkage of 0.35%. At temperature higher than 200 °C and in the heat preservation stage, a shrinkage rate similar to CS-powder-only composite and total shrinkage of 1.04% are observed, which correspond to the creep deformation of FeSiAl particles. However, in the case of CS-AMT&H$_2$O composite (indicated by red line), the shrinkage begins with the onset of pressure, reaches the first stage at 200 °C and continues in the heat preservation stage, leading to a final shrinkage of 10.67%.

High mechanical strength is the premise of engineering application. The crucial role of Al$_2$SiO$_5$ transition layer in enhancing the mechanical strength of CS-AMT&H$_2$O composite is discussed by first principle calculation. The formation energy ($\gamma$) and interaction energy ($E_{int}$) of $(110)_{FeSiAl}/(022)_{Al_2SiO_5}$ interface (Fig. 5a) are calculated to be −1.45 eV and −15.04 eV. The negative $\gamma$ indicates a favorable formation regime of interface, while the negative $E_{int}$ reveals that the slabs are strongly bonded at the interface. Figure 5b shows the three-dimensional charge density difference in CS-AMT&H$_2$O composite, where charge accumulation and charge reduction are coloured as yellow region and cyan region respectively. The charge density differences are mainly observed around the $(110)_{FeSiAl}/(022)_{Al_2SiO_5}$

interface. New Fe-O and Al-O bonds are found at the interface with lengths of 1.89 Å and 1.83 Å, respectively (Supplementary Fig. 15). The partial density of states (PDOS) in Fig. 5c and elements involved at the interface (Supplementary Fig. 15) reveal zero-band gap of the interface, suggesting an interfacial covalent bonding. The two-dimensional charge density differences in Fig. 5d illustrates that the interfacial interaction originate from the covalent nature of Fe-O and Al-O bonds. Due to the covalently bonded interface, the epitaxial Al$_2$SiO$_5$ sublayer adheres strongly to the metallic FeSiAl matrix. Moreover, compared with damaged and discontinuous FePO$_4$ coating layer in traditional CP-PA&SR composite, the interparticle space in CS-AMT&H$_2$O composite is filled with Fe$_2$(MoO$_4$)$_3$ nanocrystals. The Al$_2$SiO$_5$ layer also acts as transition layer between metallic FeSiAl matrix and interparticle oxides, provides continuous binding of neighboring particles, and endows CS-AMT&H$_2$O composite with good mechanical strength.

Since the natural oxidization layer at FeSiAl surface is amorphous, as demonstrated in Supplementary Fig. 16, we conclude that this multilayered Al$_2$SiO$_5$/SiO$_2$/Fe$_2$(MoO$_4$)$_3$ heterostructure is formed in the hydrothermal environment of cold sintering process. As revealed by the sintering shrinkage curve in Fig. 4b, AMT aqueous solution can

exist at temperature up to 200 °C, providing a hydrothermal environment in which FeSiAl surface is oxidized and dissolved. $Al_2SiO_5$ has more negative formation energy than $Al_2O_3$ and $SiO_2$ mixture, while $Fe_2(MoO_4)_3$ has more negative formation energy than $Fe_2O_3$ and $MoO_3$ mixture (Supplementary Table 2), providing the thermodynamic principle for the formation of $Al_2SiO_5/SiO_2/Fe_2(MoO_4)_3$ multilayered heterostructure. In the solvent evaporation process, $Al^{3+}$ and $Si^{4+}$ with less solubility precipitate epitaxially as $Al_2SiO_5$ transition layer on the fresh surface of FeSiAl matrix. After that, excessive $Si^{4+}$ precipitates as amorphous $SiO_2$ sublayer outside $Al_2SiO_5$. $Fe^{3+}$ and $(MoO_4)^{2+}$ with lager solubility in the solution finally crystalize as $Fe_2(MoO_4)_3$ outer layer in the anaphasis of cold sintering.

## Perspectives

Soft magnetic materials are subjected to working under alternative field. The most important and fundamental requirement in practical application is a suitable permeability at given frequency. In the past, permeability spectrum is phenomenologically related to domain wall resonance and domain rotation in multidomain/monodomain materials. Our results demonstrate that magnetic vortex, as the transitional state from monodomain to multidomain, can present frequency-stable permeability up to gigahertz frequencies when magnetically isolated. By applying ultrafine metallic magnetic particles with vortex structure, we can obtain large saturation magnetization simultaneously. The universality of frequency-stable permeability in isolated magnetic vortex can be validated by the cold-sintered composites from ultrafine Fe and amorphous FeSiCrB particles despite of their low mechanical strength. However, we believe that vortex-based SMC could also be achieved by FeSi, FeNi, FeNiMo or any other ultrafine soft magnetic particles. Meanwhile, ascribed to its high resonance frequency, we convinced that vortex-based composite also has great potentials in microwave applications such as radio frequency (RF) oscillators and spintronic devices.

In summary, we have proposed vortex-based FeSiAl soft magnetic composite with frequency-stable permeability, relatively large saturation magnetization, low coercivity and high mechanical strength. The magnetic vortexes are magnetically isolated and covalently bonded by the $Al_2SiO_5/SiO_2/Fe_2(MoO_4)_3$ multilayered heterostructure that is formed during cold sintering. The composite's permeability maintains stable up to 1 GHz with a value of 13. Meanwhile, $M_s$ of 105 $Am^2/kg$, low $H_c$ of 48 A/m and excellent ultimate compressive strength of 337.1 MPa can be achieved. Our study reveals the high frequency characteristics of magnetic vortex and sheds light on developing high-frequency magnetic devices.

## Methods

### Cold sintering of $CS$-$AMT$&$H_2O$ composite
Ultrafine FeSiAl particles with nominal composition of $Fe_{85}Si_{9.5}Al_{5.5}$ (wt %) were prepared through gas atomization and airflow classification by Hunan Hualiu New Materials Co., Ltd, China. In gas atomization process, the FeSiAl ingot is melted at 1580 °C for 20 min in nitrogen atmosphere and atomized by a pressure of 5.5 MPa. 100 g FeSiAl particles and 8 g ammonium molybdate tetrahydrate $((NH_4)_6Mo_7O_{24}\cdot4H_2O)$ were mixed with water and dried to prepare the precursor. The precursor was mixed with 20 wt% water, homogenized with a pestle and mortar, and subsequently placed in a cylinder die (diameter of 12.7 mm) and cold-sintered at 250 °C in air for 1 h under a uniaxial pressure of 400 MPa. The cold-sintered samples were machined into toroid and strip for dynamic magnetic measurements.

### Preparation of contrast samples
For CS-AMT-Fe&$H_2O$ and CS-AMT-Am&$H_2O$ composites, ultrafine ($d_{50}$ = 2.0 μm) carbonyl iron and FeSiCrB amorphous particles are also mixed homogeneously with water and AMT, and cold-sintered under the same condition to CS-AMT&$H_2O$. For CS-$H_2O$ composite, pure FeSiAl particles were mixed with 20 wt% water and conducted the above cold sintering process. For CS-powder-only composite, only FeSiAl particles were subjected to the cold sintering process. For CP-PA&SR composite, FeSiAl particles were passivated by 0.6 wt% phosphoric acid and mixed with 1.5 wt% silicon resin (SH-9602, provided by LSSH New Materials Co., Ltd, China), then pressed in a toroidal die under a uniaxial pressure of 2 GPa. For CP-PVA composite, FeSiAl particles were mixed with 8% polyvinyl alcohol and pressed in a toroidal die under a uniaxial pressure of 2 GPa.

### Characterization
The crushing strength were measured by universal testing machine (WDW-200M, ZLC) with a loading rate of 0.2 mm/min. The magnetic properties were measured by a superconducting quantum interference device (SQUID, MPMS-XL-5, Quantum Design). The permeability spectra were measured by an impedance analyzer (E4991A, Agilent). The sample was machined to $3.0 \times 1.5 \times 0.5$ mm to evaluate the performance of micro-inductors with three-turn windings. The valence states were investigated by X-ray photoelectron spectroscopy (XPS, Nexsa, ThermoFisher). The microstructure was observed by scanning electron microscopy (SEM, JSM-1T500HR, JEOL) and transmission electron microscope (TEM, JEM-ARM200F, JEOL) with a probe aberration corrector. The magnetic domain structure was observed using a TEM (Talos F200S, FEI) with differential phase contrast (DPC) mode. The elemental distribution was characterized by high-angle annular dark-field detector (HAADF) and energy dispersive X-ray detector (EDX). Before TEM observation, the composites were polished by a focused ion beam (FIB, Strata 400 S, FEI).

### First principle calculation
Calculations were done within the framework of spin-polarized density functional theory (DFT) as implemented in the Vienna ab-initio simulation package (VASP). The Perdew–Burke–Ernzerhof (PBE) version of generalized-gradient approximation (GGA) was adopted to describe the exchange–correlation interaction among electrons. Hubbard U correction was included with effective $U$ values of 4.3 eV for Fe-3$d$ orbitals and 2.0 eV for Mo-4$d$ orbitals. DFT-D2 method was used to calculate the van der Waals interactions. The plane wave cut-off was set to 420 eV. The Brillouin zone was sampled with a size-dependent G-centered k-point mesh, i.e., $7 \times 7 \times 7$ and $5 \times 5 \times 7$ for the primitive cells of FeSiAl and $Al_2SiO_5$, $7 \times 5 \times 1$ and $5 \times 4 \times 1$ for FeSiAl/$Al_2SiO_5$ simulations. The lattice parameters and atomic positions were fully relaxed until the variation of total energy was within $10^{-5}$ eV and the final force on each atom was less than 0.01 eV/Å. To simulate the interfaces, we adopted a periodic model that includes a four-layer slab of FeSiAl (110) and an adequate thickness of $Al_2SiO_5$ (022) slab respectively, with the lattice constants of 6.71 Å, 8.82 Å and 6.67 Å, 8.82 Å along a and b axes, respectively, and 15 Å thick vacuum in between slabs. The formation energy ($\gamma$) and interaction energy ($E_{int}$) of interfacial structure are calculated by formula (1) and (2):

$$\gamma = \frac{1}{A}(E_{surf}^{inter} - E_{surf}^{A} - E_{surf}^{B}) \tag{1}$$

$$E_{int} = (E_{surf}^{inter} - E_{surf}^{C} - E_{surf}^{D}) \tag{2}$$

Here, $E_{surf}^{inter}$ is the interface energy, $E_{surf}^{A}$ and $E_{surf}^{B}$ correspond to the energies of two fully relaxed surfaces, $E_{surf}^{C}$ and $E_{surf}^{D}$ represent energy of surfaces separated from the interface. In addition, we calculated the difference between the charge densities of CS-AMT&$H_2O$ composite. The charge densities are calculated using the following equation: $\Delta\rho = (\rho_{surf}^{inter} - \rho_{surf}^{A} - \rho_{surf}^{B})$, here, $\rho_{surf}^{inter}$, $\rho_{surf}^{A}$ and $\rho_{surf}^{B}$ are the charge densities of interface and corresponding individual surface slabs, respectively.

 

## Micromagnetic simulation

The micromagnetic simulation was carried out by mumax software. The cell size was varied from $0.1 \times 0.1 \times 0.1$ nm$^3$ to $50 \times 50 \times 50$ nm$^3$ depending on the simulation scale. The material parameters of FeSiAl, i.e., the saturated magnetization ($M_s$), the magnetocrystalline anisotropy constant ($K_1$), and the exchange stiffness ($A_{ex}$) were set to be 1.0 T, 0 J·m$^{-3}$, and $13 \times 10^{-12}$ J·m$^{-1}$. In the simulation of Fig. 2b, the cell size and diameter for bigger particle are 20 nm and 2560 nm respectively. In the isolated configuration, the smaller particle has a diameter of 1000 nm, and the distance between each particle is 100 nm. In the contacted configuration, the smaller particle has a diameter of 1240 nm, and it overlaps with neighboring particles by 20 nm.

## Data availability

The authors declare that the source data generated in this study are provided in the Supplementary Information and Source Data file. All source data generated during the current study are available from the corresponding authors upon request. Source data are provided with this paper.

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

## Acknowledgements

This work was supported by the National Science Fund for Distinguished Young Scholars (52225312, X.F.), the National Natural Science Foundation of China (52002103, G.H.), and the Key Research and Development Program of Zhejiang Province (2021C01193, G.H.).

## Author contributions

G.B. and X.Z. designed the experiments. G.B. and J.S. conducted material preparation. X.L. and Z.Z. performed the TEM characterization. G.B. carried out the micromagnetic simulations. W.L. and S.B. conducted the first principle calculation. Z. L. and H. Li conducted the mechanical strength measurement. N.W. prepared the PCB-embedded inductor. G.B. and X.Z. supervised and funded this work. G.B. and X.Z. wrote the initial draft and revised the manuscript. All authors participated in the discussions, contributed to improve the manuscript, and approved the submitted manuscript.

## Competing interests

The authors declare no competing interests.

## Additional information

**Supplementary information** The online version contains Supplementary Material available at https://doi.org/10.1038/s41467-024-46650-9.

