## [Peer Review File · Nature Communications]

Reviewers' Comments:

Reviewer #1:

Remarks to the Author:

The authors report about vortex-based soft magnetic composite with stable permeability. The article is clearly written and can be followed with no issues.

Below some comments:

line 14: "hundreds of MHz"

line 18: "integrated circuits"

line 39: "saturation magnetization"

line 49: reconsider a change in the sentence "... people never stop searching..."

line 55: this statement is not fully accurate: Hogan offers Fe-based powders for SMCs where particles are not spherical. Please reconsider the sentence.

line 76: can the authors explain the meaning of the term "madefaction" on the manuscript?

line 79: substitution of is per are;

line 107: why has FeSiCrB composition been selected?

line 127: The authors state that "It is clear that the crystal structure of FeSiAl particles remain unchanged during the cold sintering process" Commonly the process procedure does change crystal structure (e.g., increase of defect concentration). Do authors understand that some annealing takes place during the compaction step?

line 173: A factor of 2 different for any property is not a small mismatch. Please reconsider the sentence.

line 185: I understand that the word "facilities" should be "facilitates"?

Figure 2: the authors claim to have worked with a single particle of 150 nm, but it is not shown in Figure 2, only particle of micrometric size. It is confusing: should the image be updated?

line 363: please describe the conditions of production of sendust particles.

line 369: what is the atmosphere implemented in the process?

line 379: which silicon resin was used? How the mixing took place?

General questions (feedback could be included in the Perspectives section):

- which other materials could attend the restrictions to be a vortex-based SMC?
- which other benefits the authors could consider in developing novel vortex-based compounds?

Reviewer #3:

Remarks to the Author:

This manuscript realizes the magnetic vortex structure in magnetic particles by controlling the alloy particle size and isolating the magnetic coupling between particles. It is found that the soft magnetic composite with magnetic vortex has stable permeability and high cut-off frequency, which can be used in high frequency integrated magnetic devices. The authors have done a lot of work on both experimental and theoretical simulation, and the results support the author's conclusion. Overall, it is an innovative work and has potential application value. If the following points can be explained, it may be beneficial to improve the quality of the manuscript.

1. The thickness of the multilayer coating layer of alloy particles in the manuscript is very thin, as indicated in TEM graphs, and such insulation layer will lose its function at high frequencies, resulting in a rapid increase in eddy current losses. Does big loss affect the application of magnetic devices at high frequencies?
2. The manuscript mentions that FeSiAl particles with a magnetic vortex structure can be obtained in large quantities. In fact, FeSiAl particles with a size of less than 2.5 microns are not easy to obtain, should this statement be corrected?
3. The references are somewhat incomplete, e.g., reference 15.

Response to Reviewer 1 – NCOMMS-23-38754

Comment 1:

Line 14: "hundreds of MHz"

Line 18: "integrated circuits"

Line 39: "saturation magnetization"

Line 49: reconsider a change in the sentence "... people never stop searching..."

Response:

We are grateful that the reviewer pointed out the language errors in our manuscript. We have corrected these errors accordingly in the revised manuscript.

Modification in manuscript:

- Line 14 now reads: "Soft magnetic materials with stable permeability up to **hundreds of megahertz (MHz)** are urgently demanded in integrated transformers and inductors....."
- Line 18 now reads: ".....that are not desired in **integrated circuits**. Here, a novel frequency-stable soft magnetic....."
- Line 39 now reads: ".....i.e., large **saturation** magnetization ($M_s > 200 \text{ Am}^2/\text{kg}$) and high initial permeability....."
- Line 49 now reads: ".....induce magnetic deterioration in integrated **circuits**. **Up to now, it still remains challenging to prepare** magnetic materials with simultaneous frequency-stable permeability and large saturation magnetization."

Comment 2:

Line 55: this statement is not fully accurate: Hoganas offers Fe-based powders for SMCs where particles are not spherical. Please reconsider the sentence.

Response:

We agree with the reviewer that Hoganas offers Fe-based powders for SMCs where particles are not spherical, and statement in line 55 is not fully accurate. We have corrected the statement accordingly in the revised manuscript.

Modification in manuscript:

- Line 55 now reads: “However, present magnetic composites adopt **metallic** magnetic particles with multidomain structure”
-

Comment 3:

Line 76: can the authors explain the meaning of the term "madefaction" on the manuscript?

Line 79: substitution of is per are;

Response:

In our manuscript, term "madefaction" means wetting the powders by aqueous solution in cold-sintering process. We admit that "madefaction" is confusing, and replace it with the term “wetting” in the revised manuscript.

We have also corrected line 79 in the revised manuscript.

Modification in manuscript:

- Line 76 now reads: “The basic process involves the uniform **wetting** of powders by aqueous solution.....”
 - Line 79 now reads: “Although cold sintering techniques **are** commonly applied to fabricate.....”
-

Comment 4:

Line 107: why has FeSiCrB composition been selected?

Response:

In our work, we want to confirm if the ultrastable permeability induced by vortex structure is also applied for amorphous metallic magnetic particles. FeSiCrB composition is the only amorphous ultrafine particle that available in commercialized production. So, we choose this composition for comparison, and there is no other special reason.

Comment 5:

Line 127: The authors state that "It is clear that the crystal structure of FeSiAl particles remain unchanged during the cold sintering process" Commonly the process

procedure does change crystal structure (e.g., increase of defect concentration). Do authors understand that some annealing takes place during the compaction step?

Response:

We agree with the reviewer that some annealing takes place during the compaction step at temperature of 250°C. This annealing effect is similar to low-temperature tempering in traditional metallic materials, and changes the defect concentration in materials. In this work, FeSiAl particle size is only ~2.6 μm, much smaller than the normal grain size in bulk metallic material. Considering such small particle size, an individual FeSiAl particle is monocrystal, rather than polycrystalline. The statement in line 127 intends to convey the idea that FeSiAl particle remains monocrystal after 250 °C cold-sintering, so that the effect of grainboundary on magnetic properties can be excluded. Despite of this, we admit that our statement is a little confusing, and it will be changed in the revised manuscript.

Modification in manuscript:

➤ Line 127 now reads: “The characterizations of other particles in supplementary **Fig. 5** also demonstrate the ubiquity of monocrystal and vortex structure in the particles we adopted. Thus, the effect of grainboundary on magnetic properties can be excluded in later discussion.”

Comment 6:

Line 173: A factor of 2 different for any property is not a small mismatch. Please reconsider the sentence.

Response:

We agree with the reviewer that a factor of 2 difference for the simulated and measured coercivity is not a small mismatch. The micromagnetic simulation result represents the behavior at 0 K, which may deviate from room-temperature properties. In our work, room-temperature value (48 A/m) of CS-AMT&H₂O composite is lower than the simulated value of 91 A/m. This mismatch may be caused by the thermal perturbation at room temperature, which could decrease coercivity. Anyway, we admit that the sentence in line 173 is inappropriate and will improve it in the revised

manuscript.

Modification in manuscript:

➤ Lines 173-176 now reads: “As is well known, micromagnetic simulation represents the behaviour at 0 K, and thermal perturbation could decrease coercivity. Thus, the simulated value is higher than the room-temperature value (48 A/m) of CS-AMT&H₂O composite (shown as red star point in Fig. 2a).”

Comment 7:

Line 185: I understand that the word "facilities" should be "facilitates"?

Response:

Thank you so much for point out this language mistake! It should be "facilitates".

Modification in manuscript:

➤ Line 185 now reads: “On the contrary, isolated vortex facilitates magnetization reversal process and is beneficial for smaller H_c”

Comment 8:

Figure 2: the authors claim to have worked with a single particle of 150 nm, but it is not shown in Figure 2, only particle of micrometric size. It is confusing: should the image be updated?

Response:

In Fig. 2a of manuscript, we simulated the coercivity and magnetic structure for particles with diameters ranging from 5 nm to 10 μm , which includes the simulated properties for particle of 150 nm. However, Fig. 2b-e only deal with particles with size around 2 μm , because the averaged size (d_{50}) of FeSiAl particles in our study is $\sim 2.6 \mu\text{m}$, and most particles in the composite have micrometric size. The following Fig. R1 shows the size distribution of FeSiAl particles we used. It is easily found that 90% particles are larger than 1.3 μm . That's why we focused on particles with micrometric size in Fig. 2.

We have provided the DPC image for FeSiAl particles with size of 150 nm in supplementary Fig. 6. The DPC image is a two-dimensional projected result for three-

dimensional particle, because the particle is small enough that electron beam can penetrate through it in DPC observation. **Fig. 2e** in the manuscript shows the DPC image for FIB-thinned particles with micrometric size, because these particles are too large for electron beam to penetrate through. The DPC image for particle of 150 nm in supplementary **Fig. 6** confirms the existence of magnetic vortex structure in spherical FeSiAl particles.

HELOS (H1647) & RODOS, R2: 0.25/0.45...87.5 μ m 2022-02-18, 08:58:15_{,343}
FeSiAl

$x_{10} = 1.29 \mu\text{m}$	$x_{50} = 2.58 \mu\text{m}$	$x_{90} = 5.15 \mu\text{m}$	SMD = 2.25 μm	VMD = 2.96 μm
$x_{16} = 1.51 \mu\text{m}$	$x_{84} = 4.39 \mu\text{m}$	$x_{99} = 8.42 \mu\text{m}$	$S_v = 2.66 \text{ m}^3/\text{cm}^3$	$S_m = 9476.53 \text{ cm}^2/\text{g}$

Fig. R1. Original size distribution data of FeSiAl particles that we adopted in this work.

Supplementary Fig. 6 DPC image of FeSiAl particle with size of 150 nm shows magnetic vortex structure.

Modification in manuscript:

Line 155 now reads: “We begin illustrating the static properties of vortex-based composite by exploring a single FeSiAl particle. Because it is still challenging to observe 3D magnetic structure directly, the projected DPC image for a FeSiAl particle with size of 150 nm is taken to illustrate the vortex structure (supplementary **Fig. 6**). We also use micromagnetic simulation.....”

Comment 9:

Line 363: please describe the conditions of production of sendust particles.

Line 369: what is the atmosphere implemented in the process?

Line 379: which silicon resin was used? How the mixing took place?

Response:

In our study, we use gas atomization and airflow classification to prepare ultrafine FeSiAl particles (shown in **Fig. R2**). For gas atomization, the FeSiAl ingot is melted at 1580 °C for 20 minutes in nitrogen atmosphere. The atomization pressure is 5.5 MPa. For airflow classification, a frequency of 60 Hz is adapted. It should be noted that the preparation of FeSiAl particles is conducted in our cooperative factory (Hunan Hualiu New Materials Co., Ltd, China). We have added the information of manufacture in the revised manuscript.

The cold-sintering process in line 369 is conducted in air, and no specific atmosphere is needed.

When making the contrast sample CP-PA&SR, silicon resin SH-9602 is used (provided by LSSH New Materials Co., Ltd, China). During mixing, 1.5% silicon resin and 6% acetone are mixed homogenously with FeSiAl powders and stir-fried to completely dry.

Fig. R2. The schematic diagram of gas atomization and airflow classification

Modification in manuscript:

➤ Line 363 now reads “.....Ultrafine FeSiAl particles with nominal composition of $\text{Fe}_{85}\text{Si}_{9.5}\text{Al}_{5.5}$ (wt%) were prepared through gas atomization and airflow classification by Hunan Hualiu New Materials Co., Ltd, China. In gas atomization process, the FeSiAl ingot is melted at 1580 °C for 20 minutes in nitrogen atmosphere and atomized by a pressure of 5.5 MPa.....”

➤ Line 369 now reads “.....and cold-sintered at 250 °C in air for 1 h under a uniaxial pressure of 400 MPa.....”

➤ Line 379 now reads “.....and mixed with 1.5 wt% silicon resin (SH-9602, provided by LSSH New Materials Co., Ltd, China).....”

Comment 10:

General questions (feedback could be included in the Perspectives section):

- which other materials could attend the restrictions to be a vortex-based SMC?
- which other benefits the authors could consider in developing novel vortex-based compounds?

Response:

In our opinion, vortex structure is the transition state from monodomain to multidomain. It is the lowest-energy state for nano or micro magnetic particles considering static energy, exchange energy, demagnetizing energy. So, it is universal for all kinds of soft magnetic materials despite of their compositions. In the manuscript,

we have only validated three kinds of ultrafine particle that are available (FeSiAl, Fe, amorphous FeSiCrB). However, we believe that vortex-based SMC can also be achieved by FeSi, FeNi, FeNiMo or any other ultrafine soft magnetic particles as long as they can be prepared in large quantities.

Beyond ultrastable permeability against frequency, we think vortex-based composite have great potential in microwave applications such as radio frequency (RF) oscillators and spintronic devices due to its high resonance frequency. In these field, previous research primarily focuses on two-dimensional magnetic vortex because it can be easily fabricated and customized by thin film technology. In this work, we have demonstrated that three-dimensional magnetic vortex can also be prepared. Considering those ultrafine particles are always provided in piles, the main difficulty is the manipulation of single ultrafine particle, and perfect magnetic isolation of each ultrafine particle to maintain its vortex structure.

Modification in manuscript:

➤ The perspectives section now reads: “.....The universality of frequency-stable permeability in isolated magnetic vortex can be validated by the cold-sintered composites from ultrafine Fe and amorphous FeSiCrB particles despite of their low mechanical strength. However, we believe that vortex-based SMC could also be achieved by FeSi, FeNi, FeNiMo or any other ultrafine soft magnetic particles. Meanwhile, ascribed to its high resonance frequency, we convinced that vortex-based composite also has great potentials in microwave applications such as radio frequency (RF) oscillators and spintronic devices.

Response to Reviewer 3 – NCOMMS-23-38754

Comment 1:

The thickness of the multilayer coating layer of alloy particles in the manuscript is very thin, as indicated in TEM graphs, and such insulation layer will lose its function at high frequencies, resulting in a rapid increase in eddy current losses. Does big loss affect the application of magnetic devices at high frequencies?

Response:

We are grateful for the reviewer's comment. Big loss does affect the application of magnetic devices at high frequencies. The loss of soft magnetic materials consists of hysteresis loss, eddy-current loss and residual loss (mainly resonance loss). At high frequencies, hysteresis loss is very small because the excitation field is very low. Eddy-current loss and resonance loss dominate the total loss. The permeability spectrum is an effective way to judge the rapid increase of either eddy-current loss or resonance loss at high frequencies, since direct measurement of power loss at high frequency is still challenging today. Eddy current in soft magnetic materials will generate a reverse field to resist the excitation field, leading to rapid decline of real permeability at low frequency. In our work, the $\text{Al}_2\text{SiO}_5/\text{SiO}_2/\text{Fe}_2(\text{MoO}_4)_3$ multilayer is thicker than 20 nm. The sample CS-AMT&H₂O composite shows a very stable permeability up to GHz, signifying no rapid increase in either eddy current loss or resonance loss.

In fact, this paper tries to convey the idea that magnetic insulation is as vital as electric insulation in vortex-based SMC at high frequencies, which hasn't been revealed previously. That is to say, if two FeSiAl particles are insulated by magnetic oxide (with high resistivity), domain wall resonance will be introduced, and the permeability will decrease rapidly. To prove this, FeSiAl particles are mixed with magnetic NiFe₂O₄ nanoparticles and then cold-sintered. **Fig. R3** shows the corresponding permeability spectra. With more addition of magnetic NiFe₂O₄ nanoparticles, neighboring FeSiAl particles will be magnetically coupled. As a result, the stability of permeability spectra deteriorates rapidly with increasing NiFe₂O₄ content.

Fig. R3 Permeability spectra for FeSiAl/NiFe₂O₄ composites with different NiFe₂O₄ contents.

Comment 2:

The manuscript mentions that FeSiAl particles with a magnetic vortex structure can be obtained in large quantities. In fact, FeSiAl particles with a size of less than 2.5 microns are not easy to obtain, should this statement be corrected?

Response:

The reviewer's concern is very pertinent. A few years ago, it was indeed difficult to prepare ultrafine particles in large quantities. However, with the rapid development of powder technology, gas atomization and airflow classification are widely applied, making it possible to prepare ultrafine particles in large scale. **Fig. R2** shows the schematic diagrams of gas atomization and airflow classification, by which more than 100 kg ultrafine FeSiAl particles can be obtained per day.

Fig. R2. The schematic diagrams of gas atomization and airflow classification

Comment 3:

The references are somewhat incomplete, e.g., reference 15.

Response:

Thanks very much for your kindest reminding. We have checked again and following references have been updated.

Modification in manuscript:

➤ Reference 15, 20, 22, 23, 24 now reads:

- 15 Hrkac G, Keatley PS, Bryan MT, Butler K. Magnetic vortex oscillators. *J. Phys. D: Appl. Phys.* 48, 453001 (2015).
- 20 Watt J, et al. Gram scale synthesis of Fe/FexOy core-shell nanoparticles and their incorporation into matrix-free superparamagnetic nanocomposites. *J. Mater. Res.* 33, 2156-2167 (2018).
- 22 Mueller S, Bellaredj MLF, Davis AK, Kohl PA, Swaminathan M. Design Exploration of Package-Embedded Inductors for High-Efficiency Integrated Voltage Regulators. *IEEE Trans. Compon., Packag., Manuf. Technol.* 9, 96-106 (2019).
- 23 Bellaredj MLF, Davis AK, Kohl P, Swaminathan M. Magnetic Core Solenoid Power Inductors on Organic Substrate for System-in-Package Integrated High-Frequency Voltage Regulators. *IEEE J. Em. Sel Top. P.* 8, 2682-2695 (2020).
- 24 Ding Y, Fang X, Wu R, Sin JKO. A New Fan-Out-Package-Embedded Power Inductor Technology. *IEEE Electron Device Lett.* 41, 268-271 (2020).

Reviewers' Comments:

Reviewer #1:

Remarks to the Author:

The authors addressed the questions/suggestions proposed.

Reviewer #3:

Remarks to the Author:

The manuscript is properly revised, and I think it can be accepted.